*Method*

# Network integration of thermal proteome profiling with multi-omics data decodes PARP inhibition

Mira L Burtscher [ID] [1,2,3,4], Stephan Gade[2], Martin Garrido-Rodriguez [ID] [1,3], Anna Rutkowska[2], Thilo Werner[2], H Christian Eberl[2], Massimo Petretich [ID] [2], Natascha Knopf[2], Katharina Zirngibl[1,2], Paola Grandi [ID] [2], Giovanna Bergamini[2], Marcus Bantscheff [ID] [2], Maria Fälth-Savitski [ID] [2,5✉] & Julio Saez-Rodriguez [ID] [1,5✉]

## Abstract

**Complex disease phenotypes often span multiple molecular processes. Functional characterization of these processes can shed light on disease mechanisms and drug effects. Thermal Proteome Profiling (TPP) is a mass-spectrometry (MS) based technique assessing changes in thermal protein stability that can serve as proxies of functional protein changes. These unique insights of TPP can complement those obtained by other omics technologies. Here, we show how TPP can be integrated with phosphoproteomics and transcriptomics in a network-based approach using COSMOS, a multi-omics integration framework, to provide an integrated view of transcription factors, kinases and proteins with altered thermal stability. This allowed us to recover consequences of Poly (ADP-ribose) polymerase (PARP) inhibition in ovarian cancer cells on cell cycle and DNA damage response as well as interferon and hippo signaling. We found that TPP offers a complementary perspective to other omics data modalities, and that its integration allowed us to obtain a more complete molecular overview of PARP inhibition. We anticipate that this strategy can be used to integrate functional proteomics with other omics to study molecular processes.**

**Keywords** Proteomics; Thermal Proteome Profiling; Biological Networks; Multi-omics
**Subject Categories** Computational Biology; Proteomics; Signal Transduction

## Introduction

Cellular regulation is a complex process mediated primarily by alteration in protein activity states and abundance. These alterations can be profiled via various omics technologies. Transcriptomics can be used to estimate the activity of transcription factors. Profiling posttranslational modifications as in state-of-the-art phosphoproteomics experiments is a way to explore protein function, in particular the activity of kinases and phosphatases. Thermal proteome profiling (TPP, Savitski et al, 2014) is a mass-spectrometry (MS) based functional proteomics method that can identify shifts in protein thermal stability across different conditions (Mateus et al, 2020). Changes in thermal protein stability can reflect a series of processes that influence protein structure and its interactions independent of changes in protein abundance (Potel et al, 2021; Tan et al, 2018). Especially in combination with quantitative changes measured by e.g., phosphoproteomics, TPP can be used to identify functionally relevant alterations and link them to signaling and phenotypic consequences (Savitski et al, 2014; Mateus et al, 2020). Due to the different strengths of these methods, an integrated study would not only shed light on their complementarities, but would allow us to explore their use in a synergistic manner.

To integrate the different layers of information encoded in each of these modalities, a series of computational methods have been developed, many of them making use of biological networks as integration scaffolds (Babur et al, 2021). Depending on the network content, different types of information can be leveraged. When networks are built based on previous discoveries, this existing prior knowledge can be explored systematically (Garrido-Rodriguez et al, 2022). COSMOS is a recently published method to integrate multi-omics data and prior knowledge to extract active sub-networks in a given functional context via causal reasoning (Dugourd et al, 2021). Within COSMOS, transcription factor and kinase activities are inferred from transcriptomics and phosphoproteomics data, respectively (footprinting analysis), and then causal paths are identified that connect the enzymes coherently with their inferred activities (Dugourd and Saez-Rodriguez, 2019).

As a case study, we focused on the effect of Poly (ADP-ribose) polymerase (PARP) inhibition on DNA damage. PARPs are a family of 17 nucleoproteins, which transfer one or multiple ADP-ribose units to target proteins in a process called PARylation (Langelier et al, 2018). PARP activity has been shown to be important for a series of cellular processes, such as DNA repair, transcription, cell fate, and stress response (Gupte et al, 2017). Upon PARP inhibition, DNA damage accumulates in cells initiating a cell cycle arrest and, ultimately, apoptosis, if the damage remains unrepaired (Ray Chaudhuri and Nussenzweig, 2017).

[1]Heidelberg University, Faculty of Medicine, Heidelberg University Hospital, Institute for Computational Biomedicine, Heidelberg, Germany. [2]Cellzome, a GSK company, Heidelberg, Germany. [3]Genome Biology Unit, European Molecular Biology Laboratory, Heidelberg, Germany. [4]Faculty of Biosciences, Heidelberg University, Heidelberg, Germany. [5]These authors jointly supervised this work: Maria Fälth-Savitski, Julio Saez-Rodriguez. ✉E-mail: maria.faelth@gmail.com; pub.saez@uni-heidelberg.de

PARP inhibitors are used as tumor therapy drugs, specifically targeting cancer cells with genetic defects in DNA repair genes such as BRCA1, exploiting their synthetic lethal relationship with PARP1 (Mateo et al, 2019). Clinical applications would benefit greatly from a better understanding of the differences between certain PARP inhibitors, the influence of patients' genetic background, and potential resistance mechanisms (Mateo et al, 2019).

Here, we use a TPP dataset acquired for BRCA1-deficient ovarian cancer cells (UWB1.289) after incubation with a PARP inhibitor (Olaparib). We hypothesized that the integration of TPP data with other omics types will allow us to add context to the observed functional alterations of proteins and improve our mechanistic understanding of drug response. Toward this aim, we integrated TPP, transcriptomics, and phosphoproteomics data using the COSMOS framework in the context of PARP inhibition in ovarian cancer cells. We adapted COSMOS by modeling a signaling cascade comprising perturbed kinases as starting points, intermediate proteins with altered thermal stability and deregulated transcription factors as endpoints (Fig. 1). We found that a series of changes in thermal protein stability can be explained

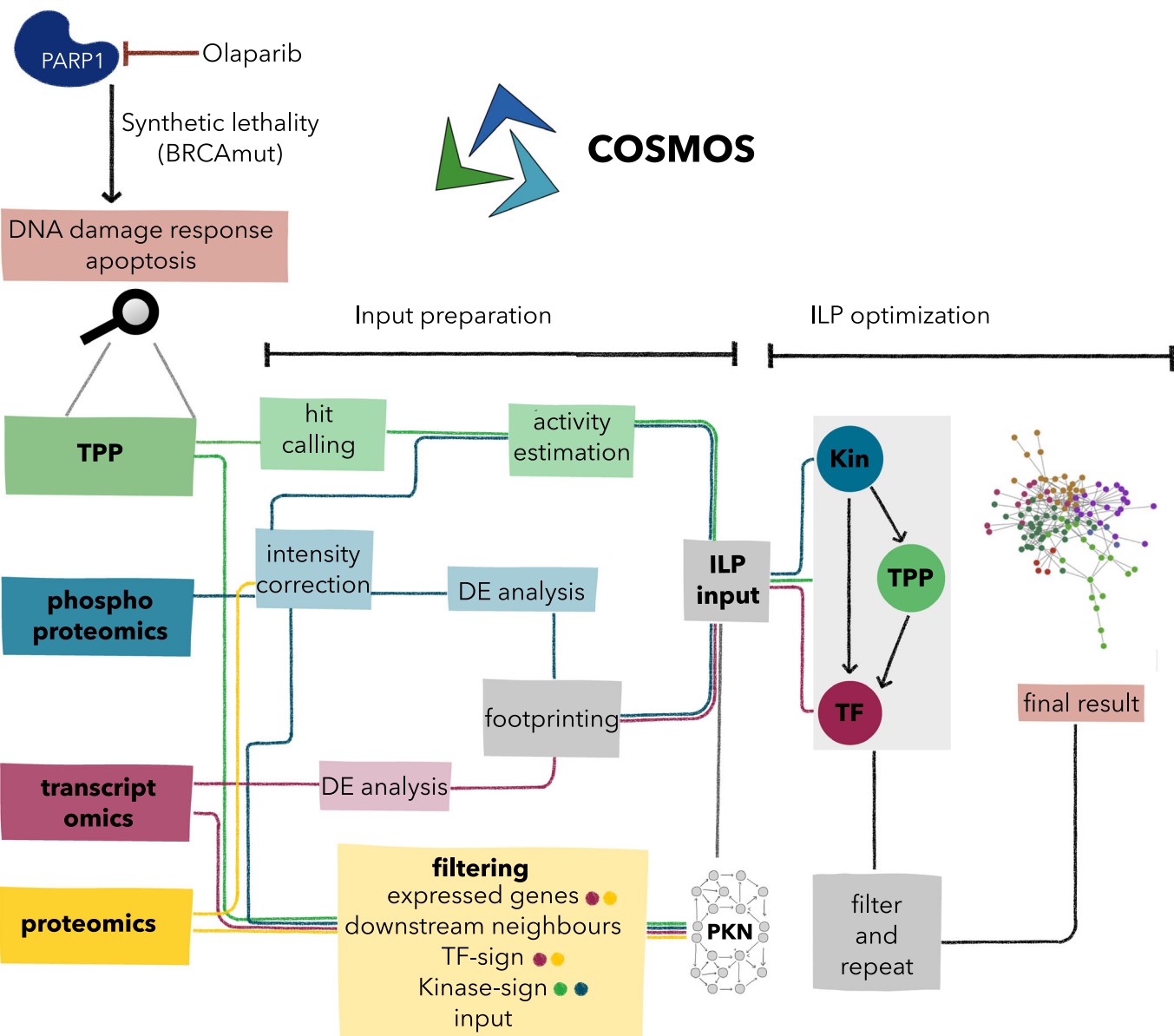

**Figure 1. Overview of the COSMOS-TPP workflow.**

Preprocessing and modeling steps of the involved data types are indicated by different colors. Olaparib is an inhibitor of PARP1, an essential player in DNA repair. Inhibition of PARP results in DNA damage that is not repaired in BRCA-mutated cells such as UWB1.289 ovarian cancer cells inducing apoptosis, a concept known as synthetic lethality. The different omic layers acquired after 24 h of Olaparib treatment are processed separately and combined as input for an integer linear programming (ILP) optimization. Additionally, different omics information is used to filter the underlying prior knowledge network (PKN). To obtain a network model describing the interplay of Kinases (Kin), TPP hits (TPP), and transcription factors (TF) two runs are merged into the final network.

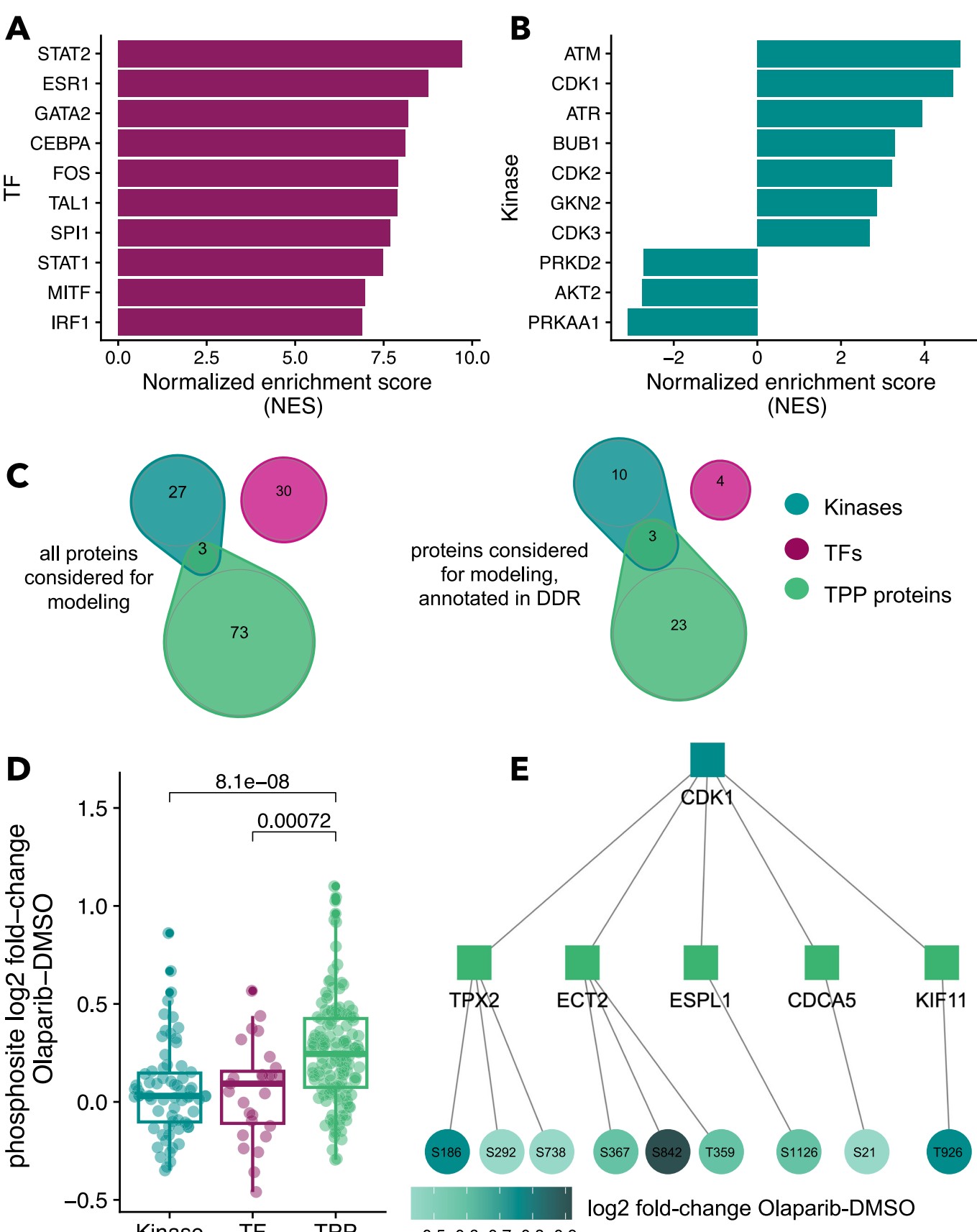

**Figure 2.  Transcriptomics, phosphoproteomics, and thermal proteome profiling data exploration.**

(A) Normalized enrichment scores (NES) for the top transcription factors and (B) top kinases as a proxy of enzyme activity based on transcriptomics and phosphoproteomics data. In response to Olaparib treatment, cells react with a strong transcriptional activation and signaling around the DNA damage checkpoint. (C) Comparison of top transcription factors, kinases, and TPP hits which were selected for network modeling overall and subsetted for proteins involved in DNA damage response (DDR). The different protein sets show almost no overlap, the three overlapping proteins between the kinase and the TPP set are CDK1, CDK2, and CCNB. (D) We compared log2 fold-changes of phosphosite intensity upon Olaparib treatment on kinases, TFs, and TPP proteins measured in the phosphoproteomics experiment. TPP proteins show a significant shift (p.value < 0.05, Wilcoxon test) towards higher phosphosite fold-changes in comparison to Kinases and TFs. $n = 244$ phosphosites in total, 70 on kinases, 26 on TFSs, and 148 on TPP proteins. Boxplots indicate the median, first, and third quartiles. Whiskers extend from the hinges to the largest value no further than 1.5× the interquartile range. Data points beyond the end of the whiskers are plotted individually. (E) TPP proteins further map frequently downstream of kinases identified in the activity estimation analysis, as exemplarily shown for CDK1 with five downstream TPP proteins and related significantly affected phosphosites.

by deregulated kinase activities and phosphoproteomic changes. Further, we showed that the integration of TPP data is essential to model and extract known and novel pathways related to PARP inhibition in detail.

# Results

## Multi-omic profiling of the response to Olaparib in ovarian cancer cells

To demonstrate the power of the proposed approach, we chose to investigate the effect of Olaparib in UWB1.289 cells. We characterized the response of UWB1.289 cells to Olaparib using transcriptomics, MS-based (phospho)proteomics, and MS-based TPP data after 24 h enabling sufficient time to generate direct and indirect response to treatment (Fig. EV1). In transcriptomics, we identified 20,493 expressed genes, from which 44 changed significantly in response to the treatment (absolute logFC > log2(1.2) and adjusted $p$ value <0.05, DESeq2). In phosphoproteomics, we identified 11,615 phosphosites, 256 of which changed their abundance significantly in response to Olaparib (absolute logFC > log2(1.2) and adjusted $p$ value <0.05, limma). In the TPP data, we identified 9455 proteins, and found 76 that suffered thermal stability changes in response to the PARPi treatment (FDR <0.1, Dataset EV 1). In this case, thermal (de)stabilization can correspond to a variety of effects ranging from drug binding to complex formation among others (Mateus et al, 2020). In summary, we generated four independent datasets that inform about different types of molecular changes happening in response to Olaparib.

## Footprint analyses reveal complementary molecular information between transcriptomics, phosphoproteomics, and thermal proteome profiling

Next, we sought to explore if regulators of the changes observed in transcriptomics and phosphoproteomics could offer complementary insight into the thermal stability changes detected by TPP. To this end, we performed a footprint analysis to estimate kinase/phosphatase and transcription factor (TF) activities from the phosphoproteomics and transcriptomics data, respectively. The main assumption behind footprint analyses is that the abundance of target transcripts of a TF can be regarded as a footprint of TF activity, and similarly for phosphosites as targets of kinases and phosphatases (Dugourd and Saez-Rodriguez, 2019, method section Footprint analyses). The analysis yielded activity estimates for 229 TFs and 170 kinases, of which 123/229 and 48/170 enzymes are considered deregulated, respectively (absolute

normalized enrichment score (absNES) >1.7). For further analysis, we focused on the top 30 significantly deregulated enzymes of each analysis to account for the most striking changes and to find an acceptable compromise of computational cost and information content for the network modeling step (Datasets EV2, EV3). The most deregulated TFs showed an increase in activity in response to the Olaparib treatment indicating transcriptional activation (Fig. 2A). Regarding affected biological processes, the top results comprised transcription factors involved in interferon (STATs, IRF1), nuclear receptor (RUNX1, ESR1), DNA repair (FOXM1, FOS/JUN) and cell cycle signaling (TFAP2C). For the phosphoproteomics data, the main deregulated kinases were found in the ATM-ATR axis, which is known to mediate DNA damage checkpoint signaling (Fig. 2B). Additionally, the downstream activation of numerous CDKs was also predicted by the footprint analysis, in line with the expected cell cycle arrest that occurs in response to PARP inhibitors. Overall, both altered transcription factor and kinase activities reflect known molecular processes affected by PARP inhibition.

Regarding the TPP data, the most prominent hits were CHEK2, PARP1, RNF146, MX1, different cyclins (stabilized), and RRN3 (destabilized). We found only a small overlap between TPP hits, transcription factors, and kinases, both across all proteins but also for proteins belonging to the DNA damage response pathway (Fig. 2C). Specifically, three proteins with relevant roles in DDR (CDK1, CDK2, CCNB1) overlapped between TPP hits and kinases, and we found no overlap between altered kinases and transcription factors. Despite the low overlap at the protein level, we found that all three omics layers inform about changes related to DNA damage response when we explored the pathways obtained from individual pathway overrepresentation analyses (Dataset EV4).

Comparison of TPP data with other omics measurements revealed 149 phosphorylation events detected on TPP hits (Dataset EV5). In detail, the TPP hits appeared to be more phosphorylated upon Olaparib treatment than kinases and transcription factors (Fig. 2D), potentially indicating the ability of TPP to capture protein activity changes upon phosphorylation. From a network perspective, 24 out of the top 30 kinases exhibit direct connections (via a single edge) to at least one TPP hit within the prior knowledge network. One example of this connection between phosphoproteomics and TPP data is CDK1, with a direct edge to 20 TPP hits (Fig. 2E). Fifty-five phosphosites which were measured in the phosphoproteomics experiment can be mapped to these TPP proteins downstream of CDK1 (examples of significantly deregulated phosphosites shown in Fig. 2E).

All of this led us to hypothesize that the three data modalities offer complementary molecular perspectives on the same biological context, and we therefore set out to apply an integrative approach to take advantage of the information they contain.

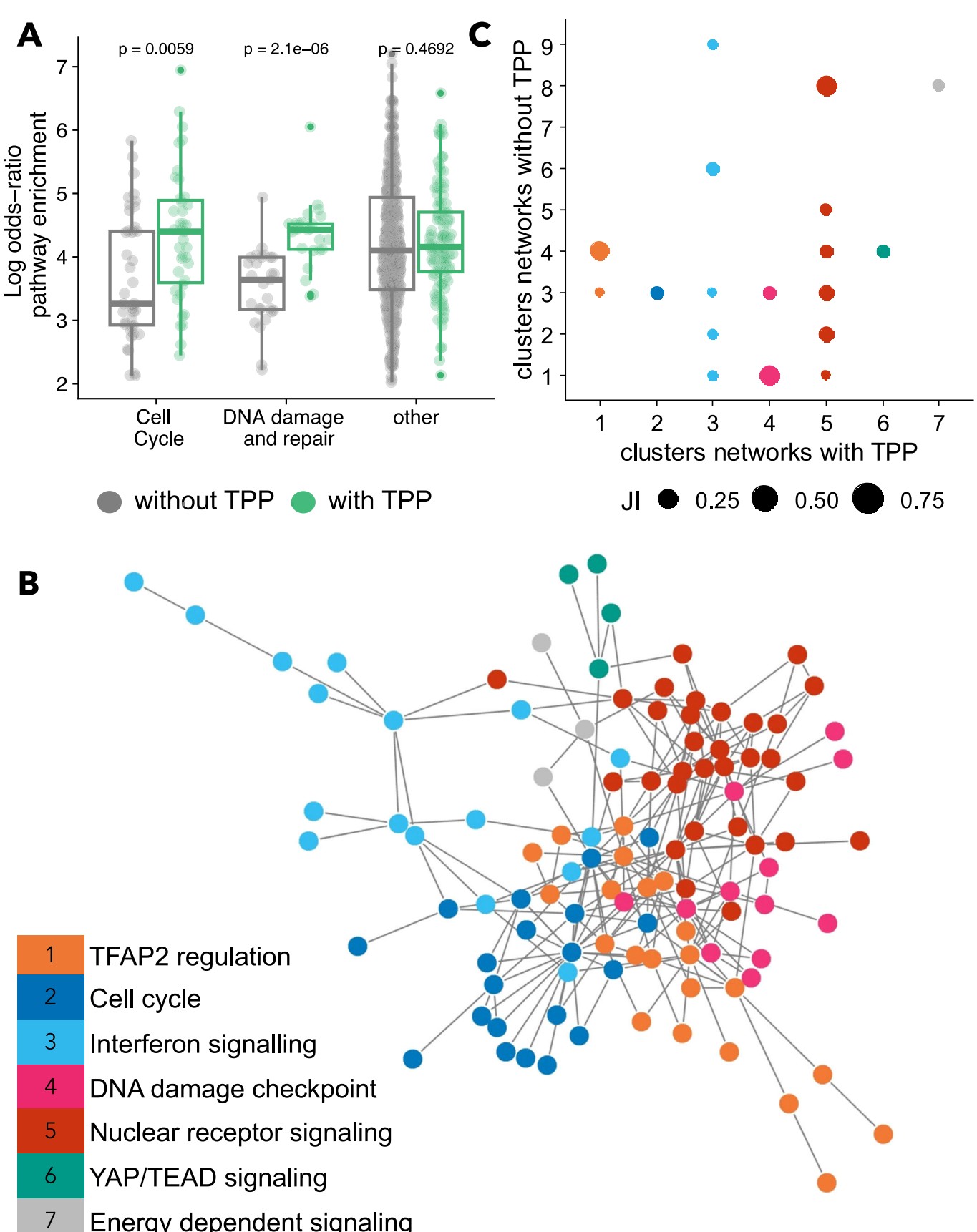

**Figure 3. Effect of TPP integration on network level.**

(A) Comparison of the network-level enrichment results of networks with TPP data and networks without TPP. The enrichment results were filtered for significant pathways (q.value < 0.05, one-tailed Fisher's exact test) and a minimum size of four nodes. We then extracted pathways related to cell cycle or DNA damage and repair and compared their log odds ratio between networks with TPP and without TPP (p.value < 0.05, Wilcoxon test). $n = 480$ pathways in total, 43 cell cycle related, 28 DDR related, 409 other pathways. Boxplots indicate the median, first, and third quartiles. Whiskers extend from the hinges to the largest value no further than 1.5× the interquartile range. Data points beyond the end of the whiskers are plotted individually. (B) Graph representation of the network with TPP data. The colors indicate the seven clusters identified using a fast and greedy clustering algorithm. We performed a cluster-specific enrichment of the nodes and summarized the significant pathways to one higher-level process per cluster ranging from cell cycle to interferon signaling. (C) We compared the overlap of TPP and no TPP clusters using the Jaccard index (JI). The colors refer to the respective cluster of the network with TPP. The size of the points refers to the Jaccard index for the given comparison. The maximum overlap reached is a Jaccard index of 0.28.

## COSMOS robustly connects kinases and transcription factors with TPP hits

To integrate the three layers of molecular data, we employed COSMOS (Dugourd et al, 2021), a recently developed method that integrates multi-omics data into causal molecular networks. To do so, it leverages the interactions that can be retrieved from prior knowledge databases, translating them into Integer Linear programming constraints. Next, it solves an optimization problem that tries to extract the smallest possible network that causally connects the maximum amount of multi-omics measurements.

In total, we used the top 30 deregulated transcription factors, top 30 deregulated kinases, and 76 significantly affected TPP proteins. In terms of network structure, we hypothesized that TPP proteins are intermediates between kinases and TFs with altered activities, as they exhibit strong phosphorylation signals that are presumably controlled by kinases and phosphatases. With the chosen setup, a TPP protein can be the downstream endpoint of kinase signaling or further connected to a TF (Fig. EV2A). We tested different COSMOS configurations to integrate the three data layers, showing that more TPP hits can be included in the result if their activity is estimated beforehand, as in the case of kinases and transcription factors (Fig. EV2B). As thermal stability changes do not inform about the relative magnitude of the deregulated activity (i.e., whether a protein activity is up or down-regulated), we estimated TPP protein activity via phosphoproteomics data when possible. For the rest of the TPP proteins, we derived their activity during optimization and supported the activity inference with a (de-)stabilization weight (Fig. EV2A). Expression information of the transcriptomics and proteomics experiments was used to further refine and tailor the network optimization (method section CARNIVAL/COSMOS analysis). This COSMOS setup yielded a maximum of integrated TPP proteins in the resulting network.

Finally, we assessed the reproducibility and robustness of solutions (Fig. EV2C). Analyses with different numbers of TPP proteins and different prior knowledge resources (Omnipath or MetaBase) showed good reproducibility between replicates, a robust clustering of networks with TPP versus networks without TPP, and an expected influence of the prior knowledge resource. With this adapted COSMOS analysis we obtained a molecular network which integrates and connects the information from the three omics layers reliably.

## Inclusion of TPP hits enhances biological information content in resulting networks

After ensuring the robustness and reproducibility of the integrated molecular networks, we aimed to profile the influence of TPP inclusion from a biological point of view. We performed an overrepresentation analysis of proteins in the networks with TPP or without TPP to identify processes which depend on the integration of TPP data to be modeled. The expected Reactome pathways around cell cycle and DNA damage were significant in both networks (q value <0.05, one-tailed Fisher's exact test, Dataset EV4) but significantly more prominent in networks with TPP (Fig. 3A, p value <0.05, Wilcoxon test). Moreover, several of these DDR-associated pathways, such as mitotic checkpoint signaling, TP53 signaling, and DNA damage recognition, were significant in the networks with TPP exclusively. This again reinforces the complementary and synergistic aspect of the TPP with respect to the other two data modalities analyzed in this study.

To extract mechanistic insights into biological processes beyond the known and well-characterized DDR, we performed a more granular analysis of different components of the networks. To this end, we performed a fast and greedy clustering of the two networks with and without TPP (Fig. 3B; Dataset EV 6). Through pathway over-representation analysis, we found specific molecular processes associated with each of the protein clusters. Top pathways per cluster ranged from expected processes like cell cycle and DDR (cluster 2 and 4) to new observations around interferon signaling (cluster 3) and YAP/TEAD signaling (cluster 6). The latter ones were not found in the enrichment analysis on a global level. In comparison, the clustering of the network without TPP yielded nine clusters with a very different composition (Fig. 3C; Dataset EV 6). The pairwise comparisons of clusters of the two network types showed a maximum Jaccard Index of 0.28 for clusters related to DDR and nuclear receptor signaling. This cluster-based network analysis allowed us to identify a series of PARP inhibition-related processes, such as DDR and interferon signaling, and highlighted the value of TPP data to model these.

To investigate the role of TPP proteins in more granular biological processes, we analyzed three pathways identified in the cluster-based enrichment (Fig. 3B). As a starting point, we focused on signaling around the DNA damage checkpoint and cell cycle arrest (Fig. 4A). These processes are well-known to happen in response to PARP inhibition and depict the biggest part of the network (Ray Chaudhuri and Nussenzweig, 2017; Masutani et al, 1995). Besides the stabilization of PARP1 as a drug target, different proteins stabilized potentially upon phosphorylation, for example, CHEK2, the major downstream target of the cell cycle arrest kinases, ATM, and ATR (Matthews et al, 2022). The network model also suggests the phosphorylation of multiple cyclins and cell cycle-regulating proteins (CCNB1, CCNB2, CCNA2, and BUB1B) by CDK1. At last, the TPP measurements also reflect complex formation as indicated exemplarily by the interaction of ESPL1 and PTTG1 which regulate the degradation of the APC/C complex during cell cycle arrest (Waizenegger et al, 2002).

Next we investigated interferon signaling, a known but less characterized response to PARP inhibition (Vikas et al, 2020; Wang

## A    Cell cycle arrest and DNA damage response

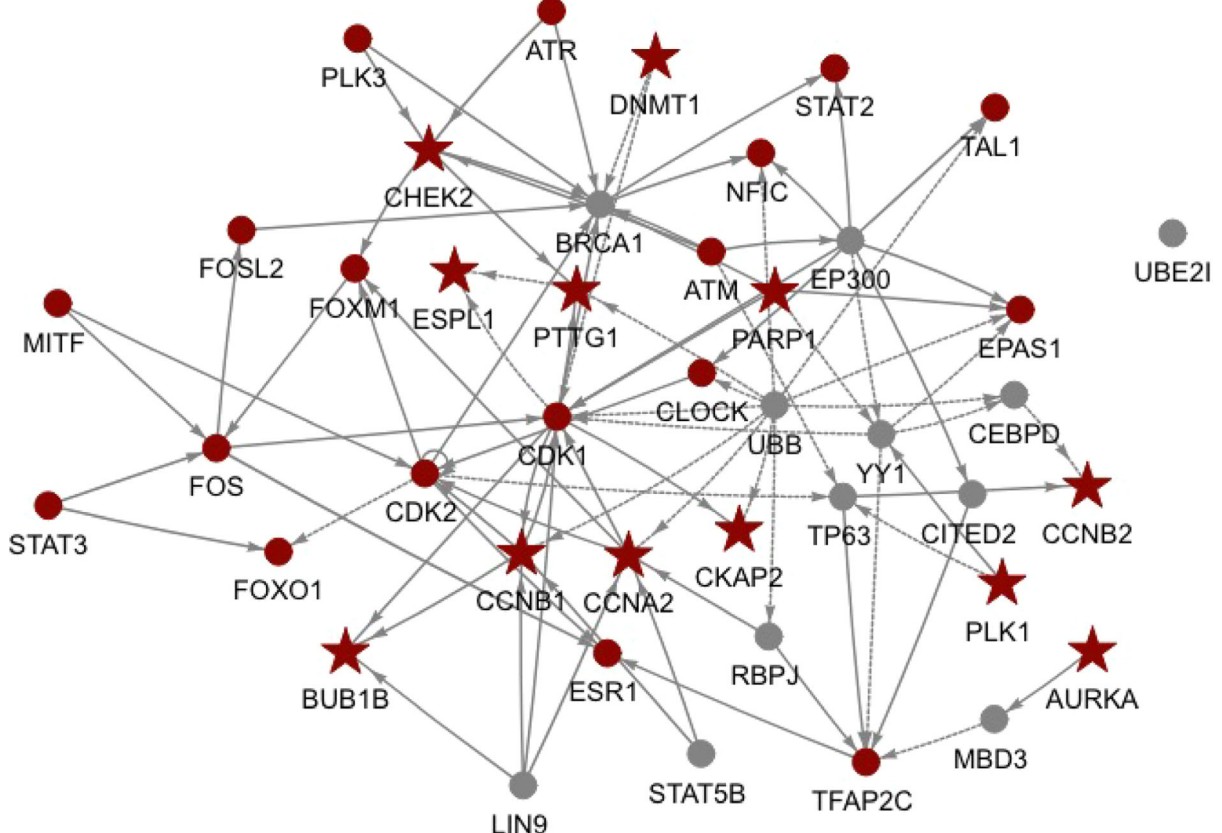

## B    Interferon signaling

## C    YAP/TEAD signaling

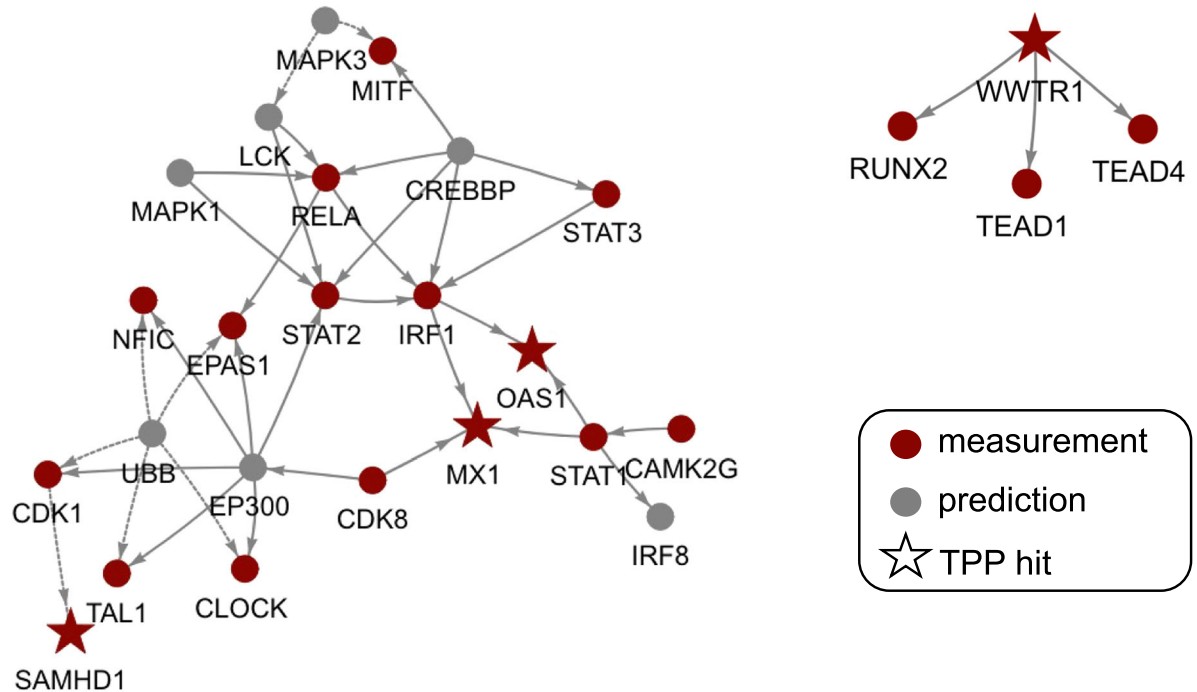

◀ **Figure 4.  Mechanistic insight into selected molecular processes.**

Based on the results of the previous clustering analysis we extracted signaling related to the identified processes from networks with TPP to investigate it in detail. The red color indicates TFs, kinases, and TPP nodes which were used as input, while gray indicates predicted nodes added from the prior knowledge network during the optimization. TPP hits are indicated as stars. We chose three pathways represented by different clusters. (**A**) DNA damage response and cell cycle arrest, (**B**) Interferon signaling, and (**C**) YAP/TEAD signaling.

et al, 2019). Interferon signaling is represented around the transcription factors, RELA, STAT1, STAT2, STAT3, and IRF1 (Fig. 4B). This transcriptional response is connected and complemented by the two TPP hits, MX1 and OAS1 which are activated upon interferon signaling (Liao and Gao, 2022; Eskildsen et al, 2003). In this case, the thermal stabilization can be linked to transcriptional changes upon perturbation with Olaparib.

Finally, we also investigated YAP-TEAD signaling, a not reported effect upon PARP inhibition (Fig. 4C), but recently described to be central in DNA damage response (Calses et al, 2023). We observe the complex of the TPP protein YAP and the TEAD1 and TEAD4 transcription factors which forms in the nucleus at the end of the hippo signaling pathway (Huh et al, 2019). The subsequently initiated transcriptional program has been shown to contribute to an aggressive and drug-resistant tumor phenotype (Kim et al, 2017).

## Discussion

In this work, we integrated thermal proteome profiling (TPP), phosphoproteomics, and transcriptomics data from cancer cell lines subjected to drug-induced DNA damage using a causal-network approach. Overall, our results demonstrate that each individual omic profile carries a signal about the response of UWB1.289 cells to the PARP inhibitor Olaparib. A detailed footprint analysis showed that the three datasets could provide complementary insights into the molecular phenomena occurring after PARP inhibition. We have therefore created a molecular network that integrates these, using our COSMOS framework (Dugourd et al, 2021).

PARP inhibitors represent a success story of modern precision medicine, with numerous approved compounds for various cancers in less than 20 years after their discovery (Mateo et al, 2019). Studies focusing on the effects of PARP inhibition at the transcriptome, proteome, or phosphoproteome level have been widely used to uncover synergies with other drugs, detect off-target effects, and characterize the pharmacological properties of various inhibitors, among others (Gupte et al, 2017; Stolzenburg et al, 2022; Perez et al, 2022; Palve et al, 2022). Here we combine, within a network model, the information of these data modalities with TPP data. Our analysis helped us determine that the inclusion of TPP results contributed to (1) Capture known biological processes related to PARP inhibition and (2) Propose new hypotheses about lesser-known mechanisms of action.

TPP has been used successfully in the past for drug profiling, but it generates complex data that are difficult to interpret and potentially benefit from integration with other data types (Savitski et al, 2014; Perrin et al, 2020; Mateus et al, 2020). By comparing the functional information provided by TPP with protein activity information derived from transcriptomics and phosphoproteomics data, we found that they complement each other in describing biological processes. The inferred transcription factors and kinases indicate the well-annotated players in the DNA damage response,

whereas proteins with altered thermal stability reflect less annotated parts of the signaling space with strong links to the phosphoproteomic measurements which aligns with observations linking functionality of phosphosites with thermal stability (Ochoa et al, 2020; Potel et al, 2021).

To characterize such links and integrate the different data modalities, we used COSMOS (Dugourd et al, 2021), that has been developed to model functional states of proteins (e.g., activities) which are more informative than protein abundance as shown before (Szalai and Saez-Rodriguez, 2020; Dugourd and Saez-Rodriguez, 2019). We show that we can integrate TPP data with transcriptomics and phosphoproteomics data and that this integration greatly improves the relevance of the resulting networks. At both the global and cluster level of the obtained networks, the inclusion of TPP enhances the reflected biological insights into PARP inhibition.

In particular, the inclusion of TPP allows us to retrieve detailed mechanistic insight into well-known consequences of PARP inhibition, such as DNA damage signaling via the ATM-ATR-CHEK2 axis or cell cycle arrest via modulation of cyclin degradation or the formation of the ESPL1-PTTG1 complex (Matthews et al, 2022; Waizenegger et al, 2002). In addition, we detect the induction of interferon signaling at the transcriptional level as well as less characterized effects such as signaling through the YAP1-TEAD complex indicated by a TPP and transcriptional signal (Huh et al, 2019; Calses et al, 2023; Vikas et al, 2020; Wang et al, 2019).

This study is the first example of the potential of TPP in network-based multi-omics studies. More comprehensive datasets covering additional contexts and data modalities would be needed to draw conclusions about the generalizability of this approach. Given the advances in mass-spectrometry technology and throughput capacity, we anticipate that the number of studies generating functional proteomics datasets such as TPP will increase. We believe that generalizing this framework and extending it to additional data types, such as solubility profiling or subcellular proteomics will be a useful step forward. Moreover, TPP data often contain more than just binary stabilization/destabilization information, such as effect size or quantitative drug response curves, which could be used to improve modeling assumptions or downstream interpretation of the network (Kurzawa et al, 2023). The interpretation of functional proteomics data could benefit further from including transcript and protein abundance information in combination with the posttranslational modification data to decipher modes of, for example, complex formation regulation.

In summary, we believe that the use of COSMOS to analyze TPP and other omics can help to extract mechanistic insight from these complex datasets.

## Methods

Reagents and tools table.

| Reagent/resource | Reference/source | Identifier/catalog number |
|---|---|---|
| **Eperimental models** | | |
| UWB1.289 cells (H.sapiens) | ATCC | CRL-2945, female |
| **Chemicals, enzymes and other reagents** | | |
| RPMI:MEGM | Lonza | #CC-3151 |
| Fetal bovine serum (FBS) | Gibco | 10270 |
| MEGM SingleQuots supplements | Lonza | #CC-3151 |
| Olaparib | AstraZeneca | CAS: 763113-22-0 |
| Dimethylsulfoxide (DMSO) | Sigma | CAS: 67-68-5 |
| Tri reagent | Thermo Fisher | #AM9738 |
| RNeasy 96-well plate kit | Quiagen | #74181 |
| NEBNext Ultra II Kit stranded | NEB | #E7760S |
| NEBNext Poly(A) mRNA Magnetic Isolation Module | NEB | #E7490 |
| SeraMag Speed beads | GE Healthcare | #45152105050250,#651521050502 |
| C18-cartridges for AssayMap | Agilent | 5190-6532 |
| Fe(III)-cartridges for AssayMap | Agilent | 5496-60085 |
| TMT reagents | Thermo Fisher | N/A |
| C18, 1.9 µm, Reprosil-Pur | Dr. Maisch | N/A |
| **Equipment** | | |
| Casy cell counter | OMNI Life Science | |
| Fragment Analyzer | Agilent | |
| Illumina NextSeq2000 | Illumina | |
| BRAVO Assaymap | Agilent | |
| Ultimate3000 nanoRLSC | Dionex | |
| Orbitrap Fusion Lumos | Thermo Fisher | |
| QExactive Plus | Thermo Fisher | |
| **Software** | | |
| Bcl2fastq v 2.20 | Illumina, Inc. | |
| Mascot 2.5 | Matrix Science, Boston, MA | |
| Trimmomatic | http://www.usadellab.org/cms/?page=trimmomatic | |
| STAR v. 1.3.4 | https://github.com/alexdobin/STAR | |
| Picard MarkDuplicates v. 2.1.1 | http://broadinstitute.github.io/picard | |
| Dorothea | https://saezlab.github.io/dorothea/ | |
| Ominpath | https://omnipathdb.org/ | |
| CosmosR | https://github.com/saezlab/cosmosR | |
| Carnival | https://saezlab.github.io/CARNIVAL/ | |
| R | version 4.2.2 | |
| RStudio | version 2022.12.0 + 353 | |
| tidyverse | version 2.0.0 | |

## Methods and protocols

### Overview of COSMOS with TPP

Here we provide a step-by-step overview of our workflow (Fig. 1). All steps are described in detail below.

Experimental setup and data generation

- **Cell culture and drug treatment**
  Ovarian cancer cells were treated with 4 uM Olaparib or a DMSO control for 24 h.
- **Data generation (following standard protocols)**

Transcriptomics data for treated and control samples were generated using next-generation sequencing. Proteomics,

phosphoproteomics, and TPP data were generated using quantitative mass-spectrometry. All data were preprocessed using the standard protocols described below.

### Computational analysis workflow

- **Hit calling**

    Differential expression analysis of treatment and control samples was performed using DESeq2 (Love et al, 2014) for transcriptomics data and using limma (Smyth, 2005) for proteomics and phosphoproteomics data. Default parameters were used if not described otherwise below. For the TPP data, hits were determined using an F-statistic-based approach (Kurzawa et al, 2020) comparing treatment and control drug-response curves.

- **Footprinting**

    For the transcriptomics and phosphoproteomics data, enzyme activities were determined using prior knowledge databases (Dorothea for transcription factor, OmnipathR for kinases and phosphatases). The Viper algorithm was used to infer activities.

- **TPP activity estimation**

    To include TPP proteins with an activity estimate in the network-based optimization thermal stability information was integrated with phosphorylation data as described in detail below. For TPP proteins without phosphorylation information we assigned a weight for the optimization based on thermal (de) stabilization.

- **Prior knowledge preprocessing**

    Two multi-omics prior knowledge networks available from COSMOS (Dugourd et al, 2021) and MetaBase (Bolser et al, 2012) were used in this study. Before optimization, the network was filtered for expressed nodes based on the measured transcriptomics and proteomics data. Further, nodes that were more than seven steps away from an input node (TFs, kinases/phosphatases, and TPP proteins) were removed. A sign filtering step to remove incoherent edges between nodes based on enzyme activity and target abundance change was applied next. Finally, input nodes that were not present in the prior knowledge network were removed.

- **Optimization**

    Having a set of input nodes of interest (TF, kinases, TPP proteins) and a preprocessed prior knowledge network, a network optimization was run using CARNIVAL (Liu et al, 2019).

    **Round 1:** To generate a first network, we used kinases as upstream input and TFs and TPP proteins as downstream measurements.

    **Round 2:** To complete the network, we performed a second round of optimization using kinases and TPP proteins as upstream input and TFs as downstream measurements

    **Postprocessing:** For postprocessing, incoherent edges (based on assigned sign and measured abundance/activity) are removed from the prior knowledge network, and both runs are then repeated to retrieve a final network

- **Result interpretation:** For biological interpretation of the result, both networks are combined into one, and duplicated edges are removed. For pathway enrichment analyses, either all nodes of the networks are used or subsets corresponding to e.g., up/downregulated nodes or certain clusters as described below.

### Cell culture and olaparib treatment

The UWB1.289 (CRL-2945; female) cell line was accessed from internal GSK collections and was obtained from ATCC. The cell line was additionally authenticated using the Promega Cell ID system. The generated short tandem repeat (STR) profiles matched exactly the expected STR profiles of the ATCC lines. Reagents were obtained from Gibco unless stated otherwise. Cells were cultured at 37 °C, 5% $CO_2$ in 1:1 RPMI:MEGM (Lonza, #CC-3151), 3% fetal bovine serum (FBS, Gibco), MEGM SingleQuots supplements (Lonza, #CC-4136) used without gentamycin-amphotericin. For the Olaparib (CAS: 763113-22-0) treatment, three 15 cm dishes per condition were prepared by seeding 3–$4 \times 10^6$ cells per plate; 24 h later medium was removed and 25 ml fresh medium containing dimethylsulfoxide (DMSO, Sigma) or 4uM Olaparib was applied to the cells. Cells were incubated for the indicated time (24 h) at 37 °C, 5% $CO_2$. Cells were collected by trypsinization, washed twice with PBS, and counted using a Casy Cell Counter (OMNI Life Science). Cell pellets were generated containing 1–2 million cells for either proteomic and phosphoproteomic or transcriptomic analysis. Three independent replicates were generated per data point per readout.

### Transcriptomics

Approximately $1 \times 10^6$ UWB1.289 were lysed in 650 µl Tri reagent (Thermo Fisher, #AM9738) and bead milled with the following settings: 4 °C, two cycles, 50% duty-cycle, 20 s, 4 m/s. RNA was extracted using the RNeasy 96-well plate kit (Qiagen, #74181) according to the manufacturer's instructions, including the DNase step. RNA concentration and integrity were assessed using a Fragment Analyzer (Agilent). Libraries were prepared using the NEBNext Ultra II Kit stranded (NEB, #E7760S) and NEBNext Poly(A) mRNA Magnetic Isolation Module (NEB, #E7490) following the manufacturer's specifications using the following options: 600 ng of total RNA per sample (starting material), poly(A) enrichment (mRNA isolation), eight PCR cycles, 10 min fragmentation time. DNA concentration and libraries size distribution were determined on a Fragment Analyzer. Libraries were pooled to 10 nM and sequenced on an Illumina NextSeq2000 instrument following the manufacturer's specifications and aiming for ~$40 \times 10^6$ reads per library. FastQ files were generated using the software bcl2fastq (version 2.20).

### Proteomics and phosphoproteomics sample preparation

Cells were lysed in 4% SDS, DNA was digested by benzonase following dilution to 1% SDS. Lysates were cleared by centrifugation and the supernatant snap frozen until further processing. All samples were processed through a modified version of the single-pot solid-phase sample preparation (SP3) protocol as described previously (Werner et al, 2021). Briefly, proteins in 2% SDS were bound to paramagnetic beads (SeraMag Speed beads, GE Healthcare,#45152105050250,#651521050502) by addition of ethanol to a final concentration of 50%. Contaminants were removed by washing 4 times with 70% ethanol. Proteins were digested by resuspending in 0.1 mM HEPES (pH 8.5) containing TCEP, chloracetamide, trypsin, and LysC following o/n incubation. Derived peptides were subjected to TMT labeling. The labeling reaction was performed in 100 mM HEPES (pH 8.5) 50% Acetonitrile at 22 °C and quenched with hydroxylamine. Labeled peptide extracts were combined into a single sample per experiment.

For phosphopeptide enrichment, parallel samples were prepared as described above. Further sample preparation was performed on the BRAVO Assaymap (Agilent Technologies). Samples were desalted using C18-cartridges for AssayMap (Agilent Technologies) according to the software protocol provided by the manufacturer. In brief, samples were

dissolved in 90 µl 6% TFA and loaded onto C18 columns equilibrated with 0.1% TFA. After loading, columns were washed with 0.1% TFA, followed by elution in 80% acetonitrile with 0.1% TFA. For phosphopeptide enrichment, Fe(III)-cartridges for AssayMap (Agilent) were used on the BRAVO Assaymap (Agilent Technologies). Cartridges were primed with 50% ACN,0,1% TFA, and equilibrated with 80% ACN, 0.1% TFA. 170 µl 80% acetonitrile, and 0.1% TFA were added to eluates from the C18 desalting step, and samples were loaded onto the Fe(III)-cartridges. Loaded Fe(III)-cartridges were washed with 80% acetonitrile, 0.1% TFA and phosphopeptides were eluted with 5% NH3 in water.

### Thermal proteome profiling

Thermal proteome profiling was performed in live UWB1.289 cells in a 2D-TPP setting as described before (Savitski et al, 2014; Zinn et al, 2021). In brief, cells were treated with four different Olaparib concentrations (0.4, 1, 4, 10 µM, or DMSO control, (one replicate per treatment) and incubated at 37 °C and 5% $CO_2$ for 24 h, then harvested by trypsinization and centrifugation. Cells were resuspended in PBS and transferred to 96-well polymerase chain reaction (PCR) plates. Cells were heated for 3 min to one of the 12 tested temperatures (42.1, 44.1, 46.2, 48.1, 50.4, 51.9, 54, 56.1, 58.2, 60.1, 62.4, and 63.9 °C). Cells were lysed with Igepal CA-630 0.8%, $MgCl_2$ 1.5 mM, and benzonase 1 kU ml$^{-1}$, and the aggregated proteins were removed by centrifugation through 0.45 µm filter plates. All flow-throughs from two adjacent temperature treatments were combined into a multiplexed TMT10 experiment. The database search was performed as described before (Savitski et al, 2014).

### Mass-spectrometry (MS) analysis

All proteomic experiments utilized TMT for relative quantification. Measurements and analyses were performed as described before (Zinn et al, 2021). In brief, expression proteomics and TPP samples were offline prefractionated using high pH reversed-phase chromatography into 8 to 24 individual fractions prior to LC-MS/MS using an Ultimate3000 RLSC (Dionex), followed by lyophilization. For expression proteomics, 12 fractions were measured. For TPP analysis, the number of fractions measured was adjusted by the expected complexity of the samples (higher temperature samples have a lower complexity): 13 fractions were measured of TMT samples containing samples derived from 42.1/44.1 °C, 46.2/48.1 °C, 50.4/51.9 °C, and 54/56.1 °C; 11 fractions were measured of the TMT sample derived from 58.2/60.1 °C and 8 fractions were measured of the TMT sample derived from 62.4/63.9 °C.

Phosphopeptide-enriched samples were measured without fractionation three consecutive times. Lyophilized samples were resuspended in 0.05% trifluoroacetic acid in water and injected into an Ultimate3000 nanoRLSC (Dionex) online coupled to a mass spectrometer. Peptides were separated on custom-made 50 cm × 100 µm (ID) reversed-phase columns (C18, 1.9 µm, Reprosil-Pur, Dr. Maisch) at 55 °C. Gradient elution was performed from 2% acetonitrile to 40% acetonitrile in 0.1% formic acid and 3.5% DMSO. Gradient length was selected based on expected complexity and necessary proteome coverage.

Expression proteomics and TPP samples were measured on a QExactive Plus instrument using 120 and 65 min gradients, respectively. MS settings were: Full MS with a scan range from 375 to 1200 m/z at a resolution of 70,000 with an AGC target of 3E6 and a maximum IT of 250 ms. Data-dependent MS2 scans (loop count 10) with a fixed first mass at 100 m/z at a resolution of 17,500, ΦSDM (Kelstrup et al, 2018) was enabled with an AGC target of 2E5 and a maximum IT of 60 ms. nCE was set at 33, the isolation window at 0.4 m/z, and dynamic exclusion at 60 s. Phosphopeptide enriched samples were measured on an Orbitrap Fusion Lumos instrument using 120 min gradients. MS settings were: Master scan with a scan range from 375 to 1200 m/z at a resolution of 60,000 with an AGC target of 4E5 and a maximum IT of 50 ms. Data-dependent MS2 mode was set to cycle time with a maximum cycle time of 3 s. The first mass was set to 100, resolution at 30,000 with an AGC target of 1E6 and a maximum IT of 64 ms. The activation type was set to HCD with a collision energy of 38%. Dynamic exclusion was set to 40 s.

Mascot 2.5 (Matrix Science, Boston, MA) was used for protein identification. In the first search, 30 ppm peptide precursor mass and 30 mDa (HCD) mass tolerance for fragment ions was used for recalibration, followed by a search using a 10 ppm mass tolerance for peptide precursors and 20 mDa (HCD) mass tolerance for fragment ions. The search database consisted of a customized version of the SwissProt sequence database (SwissProt Human release December 2018, 42 423 sequences) combined with a decoy version of this database created using scripts supplied by Matrix Science. Carbamidomethylation of cysteine residues was set as a fixed modification. Methionine oxidation, N-terminal acetylation of proteins, and TMT modification of peptide N-termini and lysine were set as variable modifications. For phosphopeptide enriched samples, phosphorylation of Ser/Thr and phosphorylation of Tyr was further added as variable modification.

### Preprocessing transcriptomic data

Raw FastQ files were processed to count matrices using the cloud processing tool DNAnexus. Adapter trimming was carried out using Trimmomatic (Bolger et al, 2014). Reads were then mapped to the reference human genome (GRCh38.96) with STAR v. 1.3.4 (Dobin et al, 2013). Picard MarkDuplicates v. 2.1.1 tool was used to identify and quantify PCR duplicates (http://broadinstitute.github.io/picard). Reads were assigned to genes using the command featureCounts from the software Subread to produce count matrices. Genes were prefiltered before differential expression testing to include only genes with more than ten counts total across all samples. Statistics of differential gene expression were calculated with DESeq2 (Love et al, 2014). The resulting P values were adjusted for multiple testing. The criteria to consider a gene differentially regulated was an adjusted P value lower than 0.05 and an absolute log2 fold change in expression greater than 1.5.

### Preprocessing proteomics data

Statistical analysis and visualization of the data were performed using the statistical language R. Phosphoproteomics data were aggregated to individual phosphosites. Log2 intensities were used as a measure for phosphorylation and quantile normalized for sample-to-sample differences. Proteomics data were filtered for proteins with at least two unique quantified peptides. The log2 sum of ion intensities was used as a measure for protein abundance. These abundance measures were normalized using quantile normalization. Differential analysis was carried out using a moderated t-test implemented in the limma package (Smyth, 2005). The resulting P values were adjusted for multiple testing. A protein was considered statistically significantly different with an adjusted P value below 0.05 and log2 fold change above log2(1.5) or below log2(1/1.5).

## Footprint analysis

In order to estimate the activity of different input nodes (transcription factors (TFs), kinases, phosphatases), the Viper algorithm was used (Alvarez et al, 2016). For transcription factors, the DOROTHEA database was used to obtain TF-target interactions with a confidence level of A, B, or C (Garcia-Alonso et al, 2019). The interactions were downloaded using the OmnipathR package (Türei et al, 2016). For the Viper algorithm, fold-change values of transcripts after limma analysis were used as input. The eset parameter was set to FALSE and the minimum target set size was set to 25 targets for one TF. Viper was further used to retrieve kinase/phosphatase activities. For this, normalized phosphosite intensities were corrected for total protein changes employing a linear model with the matched expression proteomics data. These corrected intensities were then used for differential testing before footprinting analyses. Phosphosite-enzyme collections were likewise downloaded with OmnipathR. Log2 fold-changes of phosphosites were used as input on the phosphosite level. As for TFs, the eset parameter was set to FALSE and the minimum number of measured phosphosites per kinase was set to 3. Recently, a collection of methods to perform footprinting analyses in a comprehensive manner has been released (Badia-I-Mompel et al, 2022).

## TPP hit calling and activity estimation

To identify significantly (de)stabilized proteins as hits in the 2D-TPP dataset, we used a recently introduced method (Kurzawa et al, 2020). In short, a null and an alternative model were fitted per protein and condition. The null model reflects a no-change hypothesis, while the alternative model describes concentration-dependent protein abundance changes with temperature-based constraints. Both models were fitted by minimizing the sum of squared residuals and compared using an F-statistic. Significance was adjusted by a bootstrap-based false-discovery rate (FDR) procedure. A 10% FDR cutoff was used to determine significantly (de-)stabilized proteins. We evaluated the TPP hits by comparison with the other omic data types and a pathway enrichment analysis. We performed a TPP activity estimation based on causal prior knowledge links between kinases and TPP proteins. TPP hits were compared to direct upstream kinases which were also among the top 30 regulators identified in the footprinting analysis. If the activity of all upstream kinases of one TPP hit and their prior knowledge link to the TPP hit were consistent, the TPP activity was inferred from this (i.e. Kinase A is active and has an activation PKN link leading to a TPP protein, it is active as a consequence). Further, links between kinases and TPP phosphorylation sites which were not in agreement were removed from the PKN in a kinase-sign filtering step. For the COSMOS robustness analysis, additional TPP hit sets were generated using a strict approach evaluating curve fit ($R2 > 0.8$ with sigmoidal model) and effect size (logFC $>1.5$) as well as the top 100 proteins according to a stabilization score introduced by (Perrin et al, 2020).

## Pathway analysis

For all enrichment analyses we used the ReactomePA package (Gillespie et al, 2022). For the protein activity analysis results, we used the top 30 regulators as input, for the TPP hit calling results, all hits detected and, on the network level, all nodes of the resulting network. As background, we chose all genes, phosphosites, or proteins detected for the single omics datasets and all nodes from the prior knowledge network for the network-level enrichments. In the enrichPathway() function, the organism was set to "human", minGSSize was set to five and maxGSSize

was set to 500. To identify significant pathways, we filtered the result with a significance cutoff of 0.05 after multiple testing correction.

## CARNIVAL/COSMOS and prior knowledge resources

To infer signaling processes, the CARNIVAL algorithm combines upstream perturbation information (i.e., drug targets) with downstream measurements (i.e., transcription factor (TF) activities) and a causal prior knowledge network (PKN) (Liu et al, 2019). The enzyme activities are calculated using a footprinting method, as described (method section Footprint analysis; (Dugourd and Saez-Rodriguez, 2019). The PKN contains the signed and directed protein–protein interactions (node A activates/inactivates node B), which are used to model signaling interactions between measurements and perturbation input by minimizing an objective function in an integer linear programming (ILP) optimization.

In this study, we used a prior knowledge network which is provided as part of the COSMOS package (30k interactions) and a second prior knowledge network based on protein–protein interactions of MetaBase (Bolser et al, 2012) (117k interactions). The prior knowledge resources were filtered and adapted using protein expression information from transcriptomics and proteomics experiments as described by Dugourd et al (Dugourd et al, 2021). As measured input, we used the top 30 transcription factors (NES >4) and kinases/phosphatases (NES >2) from the activity estimation analysis as well as the 76 hits of the TPP dataset (FDR <0.1) to find an acceptable compromise of computational cost and information content for the network modeling step. Eleven of these TPP hits had an activity estimate, the others were used with a weight reflecting (de)stabilization.

## Setup of COSMOS with TPP

We first used kinases and TPP proteins as upstream nodes and transcription factors as downstream nodes. This represents the signaling cascade in response to a stimulus going through kinases and proteins with altered thermal stability before it causes transcriptional effects. The prior knowledge then was filtered in between runs as described by Dugourd et al before we performed a second run from kinases upstream to TPP proteins and transcription factors downstream to complete the network. With the chosen setup, the network can end with a TPP protein or a transcription factor. Both runs were repeated once after the PKN filtering to improve the result quality. We made the union of the two final networks, resulting in a combined complete network. To test whether CARNIVAL's formulation, in combination with the adaptations for multi-omics introduced with COSMOS, can be used this way, we tested its reproducibility in different settings. We generated a bigger and a smaller TPP protein set (100 and 23 hits) using alternative hit-calling strategies (Perrin et al, 2020). For all three TPP sets as well as kinases and transcription factors alone (noTPP), we produced three network replicates for each of the two prior knowledge resources. We compared the results based on the integration of nodes and their activity prediction.

## Network enrichment analysis and comparison

For the three replicates of the network with TPP hits determined using the F-statistic-based results and three network replicates without TPP, we performed a node enrichment analysis as described using ReactomePA (Gillespie et al, 2022). We compared the results of the enrichments by calculating the log odds ratio for each pathway and extracting pathways relevant to the cell cycle and

DNA damage response. A t-test was used to determine the differences between biological processes in networks with TPP and networks without TPP.

### Clustering, cluster comparison, and cluster enrichment

To retrieve a more granular view of biological processes on the network level, we performed a clustering analysis. We extracted all edges identified in one replicate for networks with TPP hits and networks without TPP hits, respectively and applied a fast and greedy clustering algorithm to the resulting graph. The pairwise Jaccard index was used to compare the composition of the resulting clusters of networks with TPP and without TPP. We performed a cluster-specific enrichment analysis as described before and extracted all nodes related to the identified processes to gain mechanistic insights.

## Data availability

The datasets and computer code produced in this study are available in the following databases: (1) Proteomics, phosphoproteomics, and TPP data: ProteomeXchange Consortium via the PRIDE partner repository, PXD046536, PXD046613, PXD046614. (2) Transcriptomics data: Gene Expression Omnibus, GSE243208. (3) Code and source data: Github, https://github.com/saezlab/COSMOS-TPP.

## Peer review information

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

## Acknowledgements

We thank Matthias Kalxdorf, Nao Iwamoto, and Aurelien Dugourd for very helpful discussions and analysis support during this project.

## Author contributions

**Mira L Burtscher**: Conceptualization; Software; Formal analysis; Investigation; Visualization; Writing—original draft; Writing—review and editing. **Stephan Gade**: Conceptualization; Software; Supervision; Writing—original draft. **Martin Garrido-Rodriguez**: Supervision; Writing—original draft; Writing—review and editing. **Anna Rutkowska**: Data curation; Investigation. **Thilo Werner**: Data curation; Investigation. **H Christian Eberl**: Data curation; Investigation. **Massimo Petretich**: Data curation; Investigation. **Natascha Knopf**: Data curation; Investigation. **Katharina Zirngibl**: Conceptualization. **Paola Grandi**: Supervision; Project administration. **Giovanna Bergamini**: Supervision; Project administration. **Marcus Bantscheff**: Supervision; Project administration. **Maria Fälth-Savitski**: Supervision; Methodology; Project administration; Writing—review and editing. **Julio Saez-Rodriguez**: Conceptualization; Supervision; Project administration; Writing—review and editing.

## Funding

## Disclosure and competing interest statement

Saez-Rodriguez J reports funding from GSK, Pfizer, and Sanofi and fees/honoraria from Travere Therapeutics, Stadapharm, Astex, Owkin, Pfizer, and Grunenthal. Garrido-Rodriguez M was supported through state funds approved by the State Parliament of Baden-Württemberg for the Innovation Campus Health + Life Science Alliance Heidelberg Mannheim. Gade S, Rutkowska A., Werner, T., Eberl H.C., Petretich M., Knopf N., Zirngibl K., Grandi, P., Bergamini G., Bantscheff M. are employees of and holds/stocks shares in GSK. Fälth-Savitski M. holds stocks in GSK. JSR is a member of the Advisory Editorial Board of Molecular Systems Biology. This has no bearing on the editorial consideration of this article for publication.

# Expanded View Figures

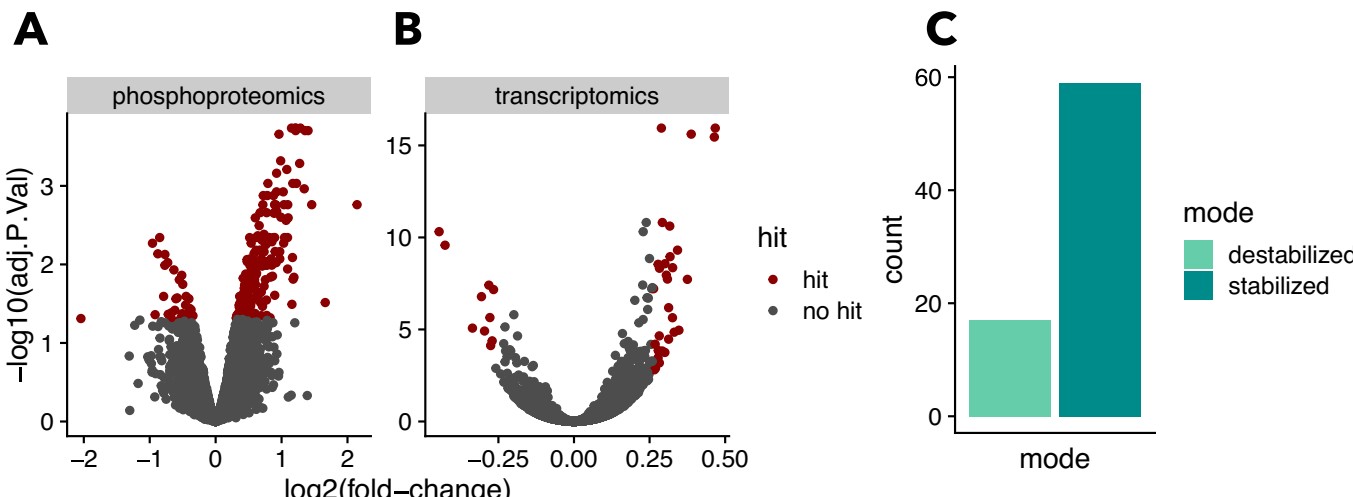

**Figure EV1. Transcriptomics, phosphoproteomics, and TPP data.**

UWB1.289 cells were treated with 4 uM Olaparib for 24 h to generate direct and indirect treatment effects before multiple omics datasets were acquired. (**A**) Volcano plot of the phosphoproteomics data after limma analysis comparing Olaparib treatment to a DMSO control ($n = 11610$ phosphosites, adjusted *p*.value <0.05, *t*-test). (**B**) Volcano plot of the transcriptomics data after limma analysis comparing Olaparib treatment to a DMSO control ($n = 13{,}710$ transcripts, adjusted *p*.value <0.05, *t*-test). (**C**) Overview of the Olaparib TPP hits determined using the F-statistic-based hit calling approach.

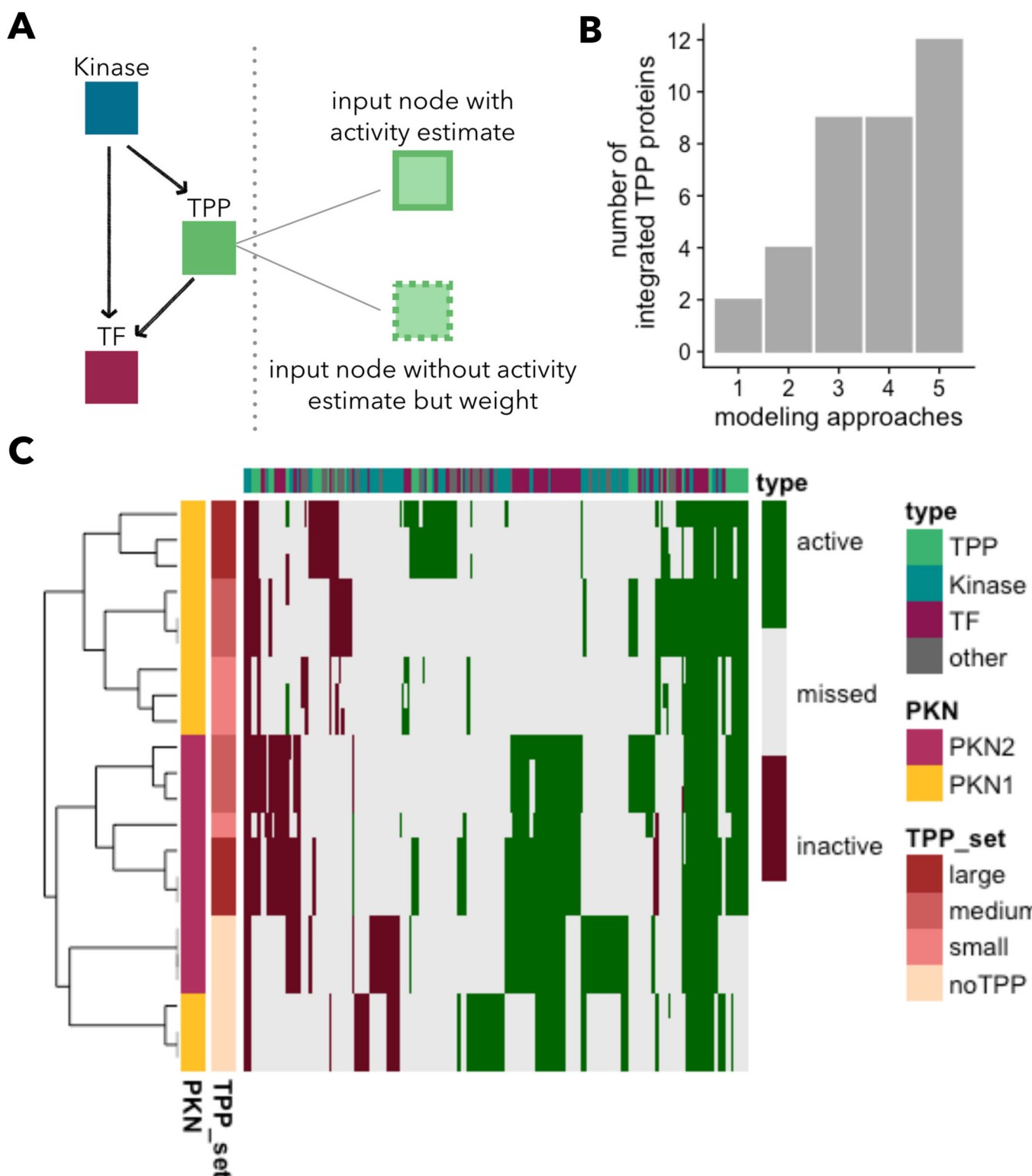

◄ **Figure EV2.  Robustness and reproducibility.**

We tried to validate whether the chosen setup (combination of runs, activity estimation and filtering steps) can be used to model reasonable networks with COSMOS. (A) We integrated TPP proteins between upstream kinases and downstream transcription factors into the signaling cascade based on the correlation of phosphoproteomic and TPP data. Further, we implemented a strategy for activity estimation where we inferred activities for TPP proteins via upstream kinases and phosphorylation status. For TPP proteins without activity estimate, we inferred the activity state during the optimization supported by a weight reflecting (de)stabilization. (B) Different modeling setups (1–5) were tested to maximize the number of integrated TPP proteins which we reached using activity estimations and weights. 1: no TPP proteins as input. 2: TPP proteins without activity information. 3: TPP with activity estimation. 4: TPP with activity estimation and weights. 5: TPP with activity estimation, NA values and weights. Running the optimization without TPP data still yields few TPP proteins in the solution, due to overlaps with other input modalities or the inclusion of prior knowledge. Details of the optimization process are provided in the CARNIVAL and COSMOS publications (Liu et al, 2019; Dugourd et al, 2021). (C) In principle, the integer linear programming solver optimizes towards a local optimum and the complete search space of the problem is not known. As a consequence, multiple technical replicates of COSMOS can have similar but slightly different optimal solutions. To assess if this solver-dependent variability can be distinguished from actual differences due to the used input data, we set up a small robustness analysis. We clustered replicate networks for different TPP set sizes based on node activities (active, missed, inactive). All replicate runs of distinct TPP sets cluster together. The networks separate into networks with TPP input and networks without TPP input. Further the influence of the used prior knowledge is clearly visible (PKN).

