## [Peer Review File · Molecular Systems Biology]

Network integration of Thermal Proteome Profiling with multi-omics data decodes PARP inhibition

Mira Burtscher, Stephan Gade, Martin Garrido-Rodriguez, Anna Rutkowska, Thilo Werner, Hans-Christian Eberl, Massimo Petretich, Natascha Knopf, Katharina Zirngibl, Paola Grandi, Giovanna Bergamini, Marcus Bantscheff, Maria Fälth-Savitski, and Julio Saez-Rodriguez

Corresponding author(s): Julio Saez-Rodriguez (pub.saez@uni-heidelberg.de) , Maria Fälth-Savitski (sara_maria.faelth_savitski@boehringer-ingenelheim.com)

Review Timeline:

Submission Date:	17th Jan 24
Editorial Decision:	22nd Jan 24
Revision Received:	13th Feb 24
Accepted:	20th Feb 24

Editor: Maria Polychronidou

Transaction Report:

The reviewers' comments and authors' responses are not available with this article, as the initial review process took place with another journal.

22nd Jan 2024

Manuscript Number: MSB-2024-12224

Title: Network integration of Thermal Proteome Profiling with multi-omics data decodes PARP inhibition

Dear Julio,

Thank you again for submitting your work to Molecular Systems Biology. I have now read your manuscript, the referee reports from the other journal and your responses to these comments and discussed them with the team. Overall, we find the reports constructive and therefore we decided to consider the study for publication using these reports, and without reviewing the study from scratch. Reviewers #1 and #2 find the presented approach (ie. the integration of Thermal Proteome Profiling (TPP) data with transcriptomics and (phospho)-proteomics data to build network models using COSMOS) a relevant contribution to the field. These two reviewers did not raise technical concerns, and mostly recommended text edits to improve clarity. Reviewer #3 was somewhat less supportive, mainly because they felt that there was a lack of clarity in the approach and justification for data analysis choices. We think that the performed revisions seem to have addressed the issues raised by all three reviewers. We also find that the inclusion of TPP data in network-based multi-omics analyses seems like a relevant contribution to the field. As such, we have decided to proceed with publishing the study in Molecular Systems Biology, pending some minor revisions, all related to editorial issues as listed below.

- We would recommend using the article type "Method", since the focus of the manuscript is more on the approach.
- Our data editors have noticed that the following information is missing in the figure legends:
 - Please define the annotated p values *******/***** in the legend of figure 3a; as appropriate.
 - Please indicate the statistical test used for data analysis in the legends of figure 2d, supplementary figures 1a-b.
 - The box plots need to be defined in terms of minima, maxima, centre, bounds of box and whiskers, and percentile in the legends of figures 2d; 3a.
 - Information related to n is missing in the legends of figures 2d; 3a, supplementary figures 1a-b.
- Please provide a .doc version of the manuscript text (including legends for main figures and tables) and individual files for the main figures. The figure legends should be included at the end of the manuscript text, after the References.
- Please note that our editorial policy does not allow "Data not shown". Please amend the statement in the legend of Figure EV1.
- Please include a Disclosure and Competing Interests Statement in the main text.
- We have replaced Supplementary Information by the Expanded View (EV format). In this case, all additional figures can be provided as EV Figures. Please provide one file per EV Figure. Their legends should be included in the manuscript text. For detailed instructions regarding expanded view please refer to our Author Guidelines: .
- Supplementary Tables S1-S6 should be provided as EV datasets (either as .xls files or .zip folders). Please provide one file per EV Dataset. Please include the description of each EV Dataset in the dataset file itself, ie. in a separate tab for .xls files or as a README.txt file in .zip folders.
- Please provide a "standfirst text" summarizing the study in one or two sentences (approximately 250 characters), three to four "bullet points" highlighting the main findings and a "synopsis image" (exactly 550px width and max 400px height, jpeg or png format) to highlight the paper on our homepage.
- All Materials and Methods need to be described in the main text. We would encourage you to use 'Structured Methods', our new Materials and Methods format. According to this format, the Material and Methods section should include a Reagents and Tools Table (listing key reagents, experimental models, software and relevant equipment and including their sources and relevant identifiers) followed by a Methods and Protocols section in which we encourage the authors to describe their methods using a step-by-step protocol format with bullet points, to facilitate the adoption of the methodologies across labs. More information on how to adhere to this format as well as downloadable templates (.doc or .xls) for the Reagents and Tools Table can be found in our author guidelines: . An example of a Method paper with Structured Methods can be found here:
- Please include a Data availability section describing how the data, code etc. generated in this study have been made available. This section needs to be formatted according to the example below:
The datasets and computer code produced in this study are available in the following databases:
 - Chip-Seq data: Gene Expression Omnibus GSE46748 (<https://www.ncbi.nlm.nih.gov/geo/query/acc.cgi?acc=GSE46748>)
 - [data type]: [full name of the resource] [accession number/identifier] [(doi or URL or identifiers.org/DATABASE:ACCESSION)]

- The Data Availability section should be reserved only for data generated in the study. Data retrieved from other sources (e.g. previously published data) should be referenced/described in a separate section titled "Datasets analysed". Code Availability should be combined with Data Availability in a single section.
- The References should be formatted according to the Molecular Systems Biology reference style (i.e., ordered alphabetically and listing the first 10 authors followed by et al).
- Molecular Systems Biology supports formal data citations in the Reference list, to cite previously published datasets. In addition to citing the original papers that reported the data, we encourage you to also cite the relevant datasets directly in the Reference list. In the text, references to datasets are included as "Data ref: Smith et al, 2001" or "Data ref: NCBI Sequence Read Archive PRJNA342805, 2017". In the Reference list, data citations are very similar to normal literature references but must be labeled with "[DATASET]" at the end of the reference. For detailed instructions please refer to our Author Guidelines .
- The funding information provided in the manuscript text should match the information entered in the online submission system.
- Please remove the 'Authors Contributions' from the manuscript. The 'Author Contributions' section is replaced by the CRediT contributor roles taxonomy to specify the contributions of each author in the journal submission system. Please use the free text box in the 'author information' section of the online submission system to provide more detailed descriptions if needed (e.g., 'X provided intracellular Ca⁺⁺ measurements in fig Y').
- When you resubmit your manuscript, please download our CHECKLIST (<https://bit.ly/EMBOPressAuthorChecklist>) and include the completed form in your submission. *Please note* that the Author Checklist will be published alongside the paper as part of the transparent process (<https://www.embopress.org/page/journal/17444292/authorguide#transparentprocess>)

Please resubmit your revised manuscript online, with a covering letter listing amendments and responses to each point raised by the referees. Please resubmit the paper ****within one month**** and ideally as soon as possible. If we do not receive the revised manuscript within this time period, the file might be closed and any subsequent resubmission would be treated as a new manuscript. Please use the Manuscript Number (above) in all correspondence.

Click on the link below to submit your revised paper.

Best wishes,

Maria

Maria Polychronidou, PhD
Senior Editor
Molecular Systems Biology

If you do choose to resubmit, please click on the link below to submit the revision online before 21st Feb 2024.

IMPORTANT:

Please note that corresponding authors are required to supply an ORCID ID for their name upon submission of a revised manuscript (EMBO Press signed a joint statement to encourage ORCID adoption). (<https://www.embopress.org/page/journal/17444292/authorguide#editorialprocess>)
Currently, our records indicate that the ORCID for your account is 0000-0002-8552-8976.

Please click the link below to modify this ORCID:
Link Not Available

***** PLEASE NOTE ***** As part of the EMBO Press transparent editorial process initiative (see our Editorial at

<https://dx.doi.org/10.1038/msb.2010.72> , Molecular Systems Biology will publish online a Review Process File to accompany accepted manuscripts. When preparing your letter of response, please be aware that in the event of acceptance, your cover letter/point-by-point document will be included as part of this File, which will be available to the scientific community. More information about this initiative is available in our Instructions to Authors. If you have any questions about this initiative, please contact the editorial office (msb@embo.org).

All editorial and formatting issues were resolved by the authors.

20th Feb 2024

Manuscript number: MSB-2024-12224R

Title: Network integration of Thermal Proteome Profiling with multi-omics data decodes PARP inhibition

Dear Julio,

Thank you again for sending us your revised manuscript. We are now satisfied with the modifications made and I am pleased to inform you that your paper has been accepted for publication.

Kind regards,

Maria

Maria Polychronidou, PhD
Senior Editor
Molecular Systems Biology
